# An esophagus cell atlas reveals dynamic rewiring during active eosinophilic esophagitis and remission

Jiarui Ding [1,2,9], John J. Garber [3,4,9] ✉, Amiko Uchida[3], Ariel Lefkovith[1], Grace T. Carter [1], Praveen Vimalathas[3,4], Lauren Canha[3], Michael Dougan[3], Kyle Staller[3], Joseph Yarze[3], Toni M. Delorey [1], Orit Rozenblatt-Rosen [1,8], Orr Ashenberg[1], Daniel B. Graham [1,4,5,6], Jacques Deguine [1], Aviv Regev [1,7,8] ✉ & Ramnik J. Xavier [1,4,5,6] ✉

Coordinated cell interactions within the esophagus maintain homeostasis, and disruption can lead to eosinophilic esophagitis (EoE), a chronic inflammatory disease with poorly understood pathogenesis. We profile 421,312 individual cells from the esophageal mucosa of 7 healthy and 15 EoE participants, revealing 60 cell subsets and functional alterations in cell states, compositions, and interactions that highlight previously unclear features of EoE. Active disease displays enrichment of *ALOX15*⁺ macrophages, *PRDM16*⁺ dendritic cells expressing the EoE risk gene *ATP10A*, and cycling mast cells, with concomitant reduction of T$_H$17 cells. Ligand–receptor expression uncovers eosinophil recruitment programs, increased fibroblast interactions in disease, and IL-9⁺IL-4⁺IL-13⁺ T$_H$2 and endothelial cells as potential mast cell interactors. Resolution of inflammation-associated signatures includes mast and CD4⁺ T$_{RM}$ cell contraction and cell type-specific downregulation of eosinophil chemoattractant, growth, and survival factors. These cellular alterations in EoE and remission advance our understanding of eosinophilic inflammation and opportunities for therapeutic intervention.

As one of the earliest points of contact between the gastrointestinal tract and a variety of environmental and dietary factors, the esophagus relies on highly coordinated interactions between multiple cell types to maintain homeostasis, and disruption of these interactions can underlie disease. In particular, eosinophilic esophagitis (EoE) arises in the setting of type 2 cytokine-driven inflammation and is associated with dysphagia and fibrostenotic complications[1]. EoE incidence and prevalence have risen rapidly over the last two decades[2], mostly in the Western world. One of the earliest and most consistent findings in EoE is increased epithelial permeability and dilated intercellular space[3–5]. Impaired mucosal integrity is thought to increase exposure to dietary and environmental allergens, further compounding the EoE inflammatory cycle. Persistent inflammation in turn leads to the development of subepithelial fibrosis, due to fibroblast activation and stimulation of collagen production via eosinophil- and mast cell-derived TGF-β[6,7]. Ultimately, the combination of structural changes

[1]Klarman Cell Observatory, Broad Institute of MIT and Harvard, Cambridge, MA 02142, USA. [2]Department of Computer Science, University of British Columbia, Vancouver, BC V6T 1Z4, Canada. [3]Gastrointestinal Division, Department of Medicine, Massachusetts General Hospital, Boston, MA 02114, USA. [4]Center for the Study of Inflammatory Bowel Disease, Massachusetts General Hospital, Harvard Medical School, Boston, MA 02114, USA. [5]Center for Computational and Integrative Biology, Massachusetts General Hospital, Harvard Medical School, Boston, MA 02114, USA. [6]Department of Molecular Biology, Massachusetts General Hospital, Harvard Medical School, Boston, MA 02114, USA. [7]Department of Biology, Massachusetts Institute of Technology, Cambridge, MA 02142, USA. [8]Present address: Genentech, South San Francisco, CA 94080, USA. [9]These authors contributed equally: Jiarui Ding, John J. Garber. ✉e-mail: garber.md@gmail.com; aviv.regev.sc@gmail.com; xavier@molbio.mgh.harvard.edu

induced by subepithelial fibrosis and the direct effects of eosinophils and mast cells on the smooth muscle in the esophagus leads to dysmotility and food impaction[8]. Although the histologic hallmark of EoE is intraepithelial eosinophils[9], mechanisms of eosinophil recruitment and activation and their precise roles in the broader type 2 inflammatory cascade remain poorly understood, including the nature of coordinated interactions between multiple immune[10–13], epithelial, and stromal cells[14].

Despite the frequent co-occurrence of EoE with other allergic diseases and the ability of dietary antigen removal to resolve inflammation[15], immunoglobulin E (IgE) appeared dispensable in an animal model of TSLP-dependent EoE[12], and anti-IgE therapy (omalizumab) failed to improve symptoms or decrease esophageal eosinophilia in a human trial[16]. Conversely, patients in the controlled trial of omalizumab had markedly increased tissue levels of IgG4, and a subsequent study showed a broad increase in esophageal immunoglobulins of all isotypes except IgE in active EoE[17]. Tissue levels of IgG4 correlated with esophageal eosinophil density, histologic severity scores, and expression of type 2 cytokines, including IL-4 and IL-13.

Recent studies investigated EoE using single-cell RNA-seq (scRNA-seq), but at a relatively modest scale. One study profiled 14,242 immune and non-immune esophageal cells, identifying 8 major cell types and 6 T cell subsets, and showed that esophageal (vs. duodenal) eosinophils upregulated proinflammatory pathway genes[18]. Cells from a pathogenic effector T helper 2 (peT$_H$2) subset were enriched for expression of arachidonic acid metabolism and eicosanoid production genes (e.g., PLA2G16, PTGS2, HPGDS, and ALOX5AP), expressed the T cell gastrointestinal and skin homing factor GPR15[19,20], and displayed evidence of clonal expansion in three of the four patients with active EoE in whom multiple TCR sequences were recovered. However, because esophageal epithelial cells were under-sampled, this work could not comprehensively dissect networks of epithelial-immune crosstalk. A smaller study profiling 1088 esophageal T cells from EoE patients showed similar results, including the presence of GATA$^+$ T$_H$2-like effector T cells, which produced high levels of IL-5 and IL-13[13], but did not detect IL-9-expressing T cells nor eosinophils, mast cells, or type 2 innate lymphoid cells (ILC2s). More recent studies, which profiled 40,297 cells from 10 individual patients[21,22] also focused on abundant cell types, such as mast and epithelial cells, each explored separately.

Here, we generate a comprehensive esophageal cell atlas of 421,312 scRNA-seq profiles from esophageal biopsies from 15 EoE patients (8 active, 7 remission) and 7 healthy participants with a normal pathology report from their esophageal biopsies. We identify 60 distinct cell subsets, with striking compositional and cell intrinsic changes in many cell types in EoE, highlighting underappreciated roles for multiple cell types beyond eosinophils in EoE pathogenesis along with therapeutic opportunities beyond eosinophil depletion. These include increased proportions of ALOX15$^+$ macrophages (induced by IL-4), plasma cells, and PRDM16$^+$ dendritic cells (DCs); induction of IL13R2$^+$ inflammatory fibroblasts that correlate with disease activity and abundance of IgG$^+$ plasma cells; a diminished T$_H$17 cell population producing fewer cytokines; and upregulation of multiple prostaglandin-related genes in rare, tissue-resident ILC2s expressing IL-5 and IL-13. Patients with active EoE, but not those in remission, have increased proportions of IgG$^+$ and IgM$^+$ plasma B cells compared to healthy participants. Cell interactions inferred from ligand–receptor expression that are specific to active EoE highlight mast cell–IL-9$^+$ T$_H$2 cell interactions and expression of SELP on activated endothelial cells, which can facilitate the recruitment of eosinophils. Finally, we identify cell type-specific expression of EoE risk genes, including NOVA1[23], expressed in esophageal fibroblasts, and ATP10A, expressed in PRDM16$^+$ DCs, which are enriched in active disease. Taken together, our work shows multiple cell types, states, and interaction networks that promote the development or associate with the resolution of eosinophilic inflammation.

## Results

### A human esophagus cell atlas

To understand cellular alterations during EoE, we analyzed 37 esophageal biopsies from 22 human donors (8 active EoE patients, 7 remission EoE patients, and 7 healthy participants on which an endoscopy was performed because of general dyspepsia, but where no pathological findings were reported), with equal numbers of males and females (Fig. 1a, b, Supplementary Fig. 1a, Supplementary Data 1, Methods). For 15 of the donors, we obtained and separately analyzed biopsies from both proximal and distal regions of the esophagus (Fig. 1a, b, Supplementary Fig. 1a). Each biopsy was processed individually with a rapid digestion protocol optimized for simultaneous recovery of epithelial, stromal, and immune cells (Methods).

We assigned 421,312 single-cell profiles to 60 prevalent subsets, spanning 93.5% of cells (393,763/421,312) and annotated by expression of distinct marker genes (Fig. 1c, d, Supplementary Fig. 1b and 2a, b, Methods), as well as another 12 rare cell subsets (1,215 cells, Supplementary Note 1). The 60 prevalent subsets comprised 4 epithelial cell subsets, 10 stromal and glial (Schwann) cell subsets, 17 myeloid (monocyte, macrophage, DC) subsets, 5 B cell subsets, 20 T/natural killer (NK) lymphocyte subsets, an erythroid cell subset, an eosinophil cell subset (expressing Charcot-Leyden Crystal (CLC) and CCR3), and two mast cell subsets (expressing tryptase α/β1 (TPSAB1) and CPA3) (Fig. 1c, d). A portion (21.6%) of mast cells also expressed chymase (CMA1, Supplementary Fig. 2c), suggesting that both tryptase$^+$chymase$^+$ mast cell (MC$_{TC}$) and tryptase$^+$chymase$^-$ mast cell (MC$_T$) subtypes exist in the esophageal mucosa[22]. Cells from each of the 60 subsets were observed in multiple donors from each of the three groups (Supplementary Fig. 1b). Cells from another 12 rare subsets were detected in only a few patient biopsies (95.5% from 2 healthy individuals) but were readily annotated, including foveolar, parietal, mucous neck, Paneth, chief, duct, enterochromaffin-like, and ghrelin- or gastrin-expressing cell subsets (Supplementary Fig. 2d). We did not observe neutrophils, which are not a feature of uncomplicated EoE[24], or basophils. Our endoscopic biopsies captured epithelia and lamina propria but were not deep enough to obtain submucosal neurons, which we previously reported in the healthy esophagus by single-nucleus RNA-seq of the esophagus muscularis[25].

Eleven of the 60 prevalent subsets consisted of proliferating cells, including cycling basal zone epithelial cells, fibroblasts, blood vascular endothelial cells (BECs), pericytes, mast cells, macrophages, DCs, plasma B cells, CD4$^+$ and CD8$^+$ T cells, and NK cells (Fig. 1d). Cell profiles from 10 cycling cell subsets (excluding cycling basal cells) formed a clear circular pattern when embedded by UMAP[26] using only cell cycle-specific genes[27] (Supplementary Fig. 2e) and had a higher number of detected expressed genes per cell compared to their non-cycling counterparts (Supplementary Fig. 1b, except for epithelial cells), as previously reported[28,29]. In addition to professional antigen-presenting cells (APCs), a subset of CD8$^+$ T cells, BECs (especially venous endothelial cells), and glial (Schwann) cells expressed major histocompatibility complex class II (MHC-II) genes (Fig. 1d), albeit at lower levels than macrophages and DCs. Because the transcripts were observed even after stringent removal of ambient RNA[30] (Supplementary Fig. 2f), these cells may represent non-conventional APCs in this setting. In the B cell compartment, only memory B cells expressed high levels of MHC-II genes.

The epithelial compartment included four subsets: quiescent basal cells expressing TSLP (an EoE risk gene), CXCL14, IL1R2, and KRT14/15; apical cells expressing KRT78 and CRNN; cycling basal cells; and suprabasal cells expressing a mix of basal, apical, and cell type-specific genes, such as SERPINB3/4 and DSC2 (Fig. 1d, Supplementary

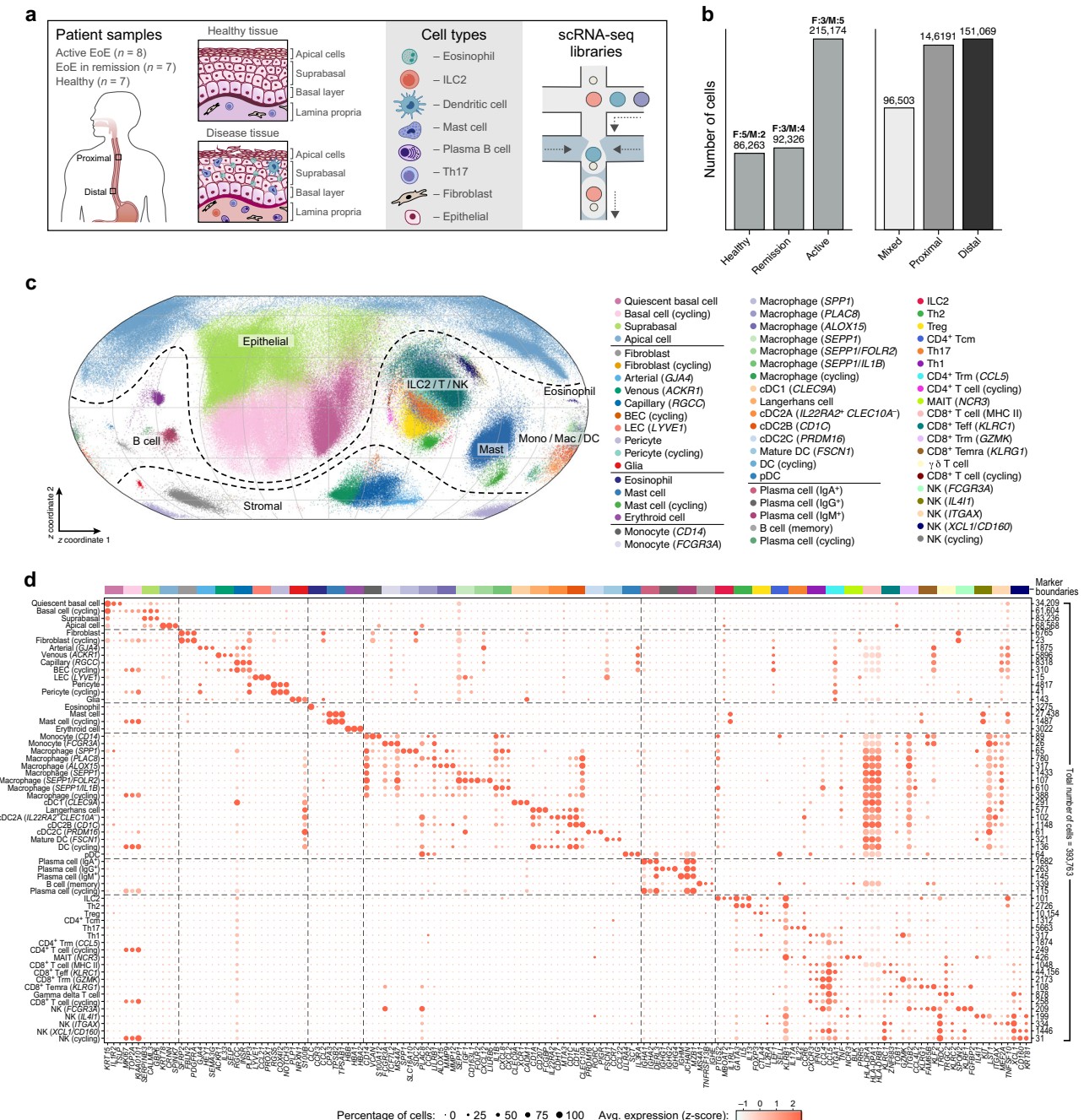

**Fig. 1 | A human esophageal cell atlas. a** Study design. **b** Data composition. Number of cells (y-axis; labeled on top of bar) in analysis from each condition (x-axis, left) or location (x-axis, right). The number of samples in each condition from males (M) and females (F) are labeled on top. Distal and proximal biopsies are paired from the same 15 donors. **c**, **d** 60 prevalent cell subsets in the esophageal atlas. **c** Two-dimensional spherical latent representation of 393,763 cell profiles (dots) from all donors (n = 22), colored by cell type, as learned by scPhere[82], taking patient, disease status, and anatomical region of biopsy as batch factors. **d** Mean expression (dot color, log(TP10K + 1), gene Z-scores across all cells of a type) and fraction of expressing cells (dot size) of marker genes (columns) for each of the 60 prevalent cell subsets (rows). Right: total number of cells of a type. Top: cell type marker genes. Source data are provided as a Source Data file.

Data 2). Our annotations of the esophageal epithelial compartment were consistent with recent studies[21,31–33], with a matching expression of marker genes (e.g., *COL17A1* in quiescent basal cells, *KRT4* in differentiated suprabasal but not basal cells, and *CRNN* mostly in apical cells[32]) (Supplementary Fig. 2g). Suprabasal cells expressed the highest level of the EoE risk gene *CCL26* (Supplementary Fig. 2h), encoding eotaxin-3, a chemotactic factor for eosinophils induced by IL-13. Cycling basal cells were partitioned into two subsets, which we termed 'differentiation' and 'renewal' (Supplementary Fig. 2i,j), by their expression of suprabasal and basal cell marker genes, respectively. A

force-directed layout embedding of a diffusion map analysis predicted that cycling 'renewal' basal cells differentiate into quiescent basal cells, while cycling 'differentiation' basal cells differentiate into suprabasal cells, which subsequently differentiate into apical cells (Supplementary Fig. 2k). In active EoE, there was reduced differentiation of suprabasal cells to apical cells, consistent with recent reports[21] (Supplementary Fig. 2k). Together, our dataset forms the most comprehensive cell atlas of the esophageal mucosa to date and allows cell subsets to be distinguished at high resolution, particularly within the lymphoid and myeloid compartments.

## Cell compositions shift in active EoE and largely restore in remission

To understand how treatments may broadly impact cellular composition within or across groups in the context of our study, we performed permutational analysis of ANOVA (PERMANOVA, Methods), with disease, age, treatment (steroids and food elimination), and use of proton pump inhibitors (PPIs) as the independent variables. Disease had a highly significant ($p = 0.00017$) impact on cell composition and steroid use was marginally significant ($p = 0.04680$). Food elimination was marginally insignificant ($p = 0.10844$). Both PPI and age were not significant ($p > 0.3$). We did an extra analysis by adding patient sex, and the results were almost unchanged, with only disease ($p = 0.00007$) and steroid ($p = 0.03829$) being significant and all the other variables, including sex, being insignificant ($p > 0.05$).

We next quantified cellular compositional changes in different disease conditions using a negative binomial regression model (Methods). Compared to both remission and health, active EoE had distinct proportions of multiple cell types beyond eosinophils, often in a manner that correlated with disease activity. Both eosinophils and mast cells increased in frequency as expected in patients with active EoE compared to remission or healthy participants (Fig. 2a, Supplementary Fig. 3a,b, Methods). Additionally, in active EoE the proportions of *PLAC8*+ macrophages, *ALOX15*+ macrophages, *PRDM16*+ DCs, and plasmacytoid DCs (pDCs) increased, while the proportions of *FOLR2*+ macrophages and apical cells decreased (Fig. 2a), highlighting additional findings in EoE. T$_H$2 cells, a major source of IL-4 and IL-13, as well as cycling CD4+ T cells and plasma B cells also increased in proportions in active EoE (Fig. 2a). Shifts in rarer cell states that were not observed in healthy participants (e.g., cycling plasma cells) could not be formally assigned statistical significance (Methods). Eosinophils, mast cells, other myeloid cells, and lymphocytes were among the top cell types whose abundances highly correlated with disease activity (absolute Spearman $\rho > 0.48$, FDR < 0.1), as measured by clinical annotation of the number of eosinophils per high-power field (HPF) in histopathology samples (Fig. 2b, c). These followed a typical gradient of cell proportions from health to remission to active EoE, although the proximal biopsies of two active EoE patients showed very few eosinophils per HPF (Fig. 2c). For patients in remission, high #eosinophils/HPF did not correlate with symptoms in our small cohort (e.g., the patient with 10 eosinophils/HPF did not have any remaining symptoms) (Supplementary Data 1). Most cells whose abundances were correlated to #eosinophils/HPF were also highly correlated with endoscopic reference scores (EREFS) of EoE (Supplementary Fig. 3c, d). Thus, increases in multiple cell types in addition to eosinophils characterize compositional changes in active EoE. Consistently, cell composition profiles distinguished active EoE from health and remission by the first principal component (PC) in a principal component analysis (PCA) based on cell proportions (Fig. 2d, Methods), with cells enriched in active EoE (eosinophils, cycling mast cells, and cycling plasma cells) highly positively correlated with PC1. Although one patient (E1054) had discordant #eosinophils/HPF in distal *vs.* proximal regions, neither the proximal nor the distal profile showed active EoE signatures. For the eight patients (excluding the two patients in active EoE but with discordant eosinophil counts in the distal and proximal regions) where both proximal and distal biopsies were separately profiled, based on a negative binomial model, we did not detect significant cellular compositional changes after correcting for multiple comparisons (FDRs ≥ 0.095).

While cellular composition was mostly restored in remission to proportions similar to those of healthy samples, there were some notable differences, where remission either retained features of active EoE or was uniquely distinctive. In particular, the significant contraction of CD4+ resident memory T (T$_{RM}$) cells in active EoE compared to health was not restored in remission (Fig. 2a, Supplementary Fig. 3b, Methods). T cell subsets, including T$_H$17 cells, CD4+ resident memory T cells, and regulatory T cells (T$_{regs}$), were among the very few cell types whose proportions differed significantly between remission and healthy participants (Fig. 2a, Methods), highlighting select features that are not simply restored in remission. These results were statistically robust. Specifically, a linear regression model to quantify cellular compositional shifts after quantile normalization of each cell type's proportion yielded largely consistent results, although with slightly lower power, (Supplementary Fig. 3c) and identified significant changes in rare cell types, such as cycling plasma cells in active EoE (Supplementary Fig. 3e). Overall, our analyses show a broad remodeling of cell composition during active disease that is largely, but not fully, resolved in remission, and includes concomitant changes in the lymphoid and myeloid compartments, which we further investigated below.

## *ALOX15*+ macrophages and *PRDM16*+ DCs associate with active EoE

Macrophages spanned multiple subsets with either tissue-resident or monocyte-derived features, including an *ALOX15*+ subset that was associated with disease. Overall, macrophages (Fig. 3a–d, Supplementary Data 3a–c) showed a predominantly M2 phenotype[34–36] (expressing *STAB1*, *CD163/209*, *F13A1*, *VSIG4*, and *TREM2*) and expressed complement component 1q genes (e.g., *C1QA*), *CD14/81*, and *MAFB*. Most macrophages expressed *SEPP1* (*SELENOP*) and spanned different states, with one upregulating *FOLR2*, *CXCL12*, and *LILRB5*, and another upregulating *IL1B* and *CXCL2/8*[37,38]. Additionally, a subset of lipid-associated *SPP1*+ macrophages[25] showed lower levels of *C1QA* and higher expression of *CD14*+ monocyte genes *S100A8/A9* and *C5AR1*, potentially originating from monocytes in the esophagus[39]. A subset of *PLAC8*+ macrophages expressed marker genes *CCR2*, *LGALS2*, and *LILRB1*, which were also expressed in monocytes and may represent recent monocyte-derived macrophages in a transient state[37]. Macrophages expressing the IL-4 polarization gene *ALOX15*+ increased in proportion in active disease (Fig. 2a) and had a similar profile to *PLAC8*+ and *SEPP1*+ macrophages, but also expressed *MMP9/12* and *TREM2*. Accordingly, cell-intrinsic expression of *ALOX15* was upregulated in cells from patients with active EoE (Fig. 3c,d). The *ALOX15*+ macrophages were characterized by an expression program identified by unsupervised non-negative matrix factorization of the macrophage subsets, including *MMP12*, *MMP9*, and *ALOX15* as the top three genes (Fig. 3e).

Esophageal DC subsets included conventional type 1 dendritic cells (cDC1s), cDC2Bs and *IL22RA2*+*CLEC10A*− cDC2As recently discovered in the spleen[40], and a subset of *PRDM16*+ cDC2Cs expressing the EoE risk gene *ATP10A* that were rare in healthy tissue and enriched in active EoE. Differential expression analysis between cDC2As and cDC2Bs recovered known cDC2A marker genes (Fig. 3f, Supplementary Data 3d). cDC2s expressing *PRDM16*, *RORC*, and *PIGR* (Fig. 3g) were enriched in active EoE (Fig. 2a). These cells did not express DC4 marker genes[41] (*FCGR3A* and *SERPINA1*) nor AS DC markers (*AXL* and *SIGLEC6*). Re-analyzing several published scRNA-seq datasets showed that cells with similar profiles are present in other tissues, including glioblastoma tumors[37] (Fig. 3h), spleen, parotid glands, lymph nodes, inguinal lymph nodes (Fig. 3i, from the cross-tissue cell atlas Tabula Sapiens[42]), mesenteric lymph nodes, lung-draining lymph nodes, lung (Fig. 3j, from cross-tissue immune cells[43]), and colon[44] (Fig. 3k). *PRDM16*+ cDC2Cs highly expressed *PTPRC*, MHC-II genes, *CD52*, *AIF1*, *LST1*, *CLEC4A*, and *TYROBP*, but not *CD68*, *MAFB*, and C1q genes, confirming them as *bone fide* DCs. Both *PRDM16*+ cDC2Cs and cDC2As did not express *CLEC10A*. *PRDM16*+ cDC2C profiles were more similar to those of cDC2As than cDC2Bs (Fig. 3a,b, based on the top four PCs), with both cDC2Cs and cDC2As expressing *ATP10A*. Some of the *PRDM16*+ cDC2Cs were in a distinct state expressing cell cycle marker genes such as *KIAA0101* and *TYMS* (Fig. 3a,b). Thus, *PRDM16*+ cDC2Cs

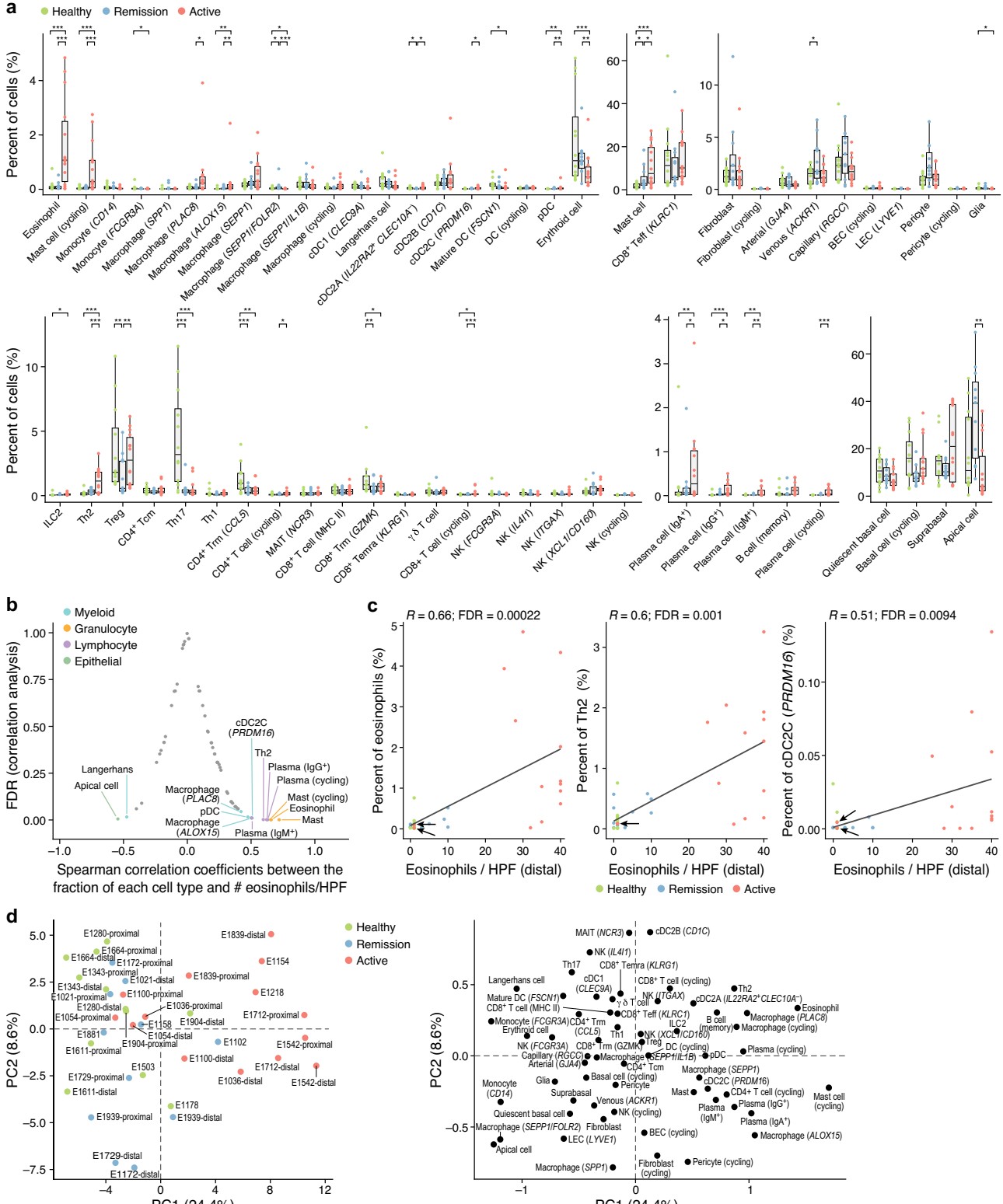

**Fig. 2 | Shifts in cellular composition during remission and active EoE.**
**a** Changes in cell composition between conditions. Distributions of cell type proportions (*y*-axis) in active disease (red, *n* = 14), remission (blue, *n* = 11), or healthy (green, *n* = 12) biopsies (points). Boxplots: medians and interquartile ranges (IQR). Whiskers: lowest datum within 1.5 IQR of the lower quartile and highest datum within 1.5 IQR of the upper quartile. ***BH FDR < 0.001, **FDR < 0.01, *FDR < 0.05, two-tailed Wald test. **b**, **c** Cell proportion association with eosinophil infiltration. **b** FDR (*y*-axis, two-tailed one-sample Student's *t* test) of Spearman rank correlation coefficients (*x*-axis) between number of eosinophils per high power field (HPF) and

the number of cells of each subset in each patient (*n* = 22). Cell types with FDR < 0.1 are shown. **c** Percent cells of specific types (*y*-axis) and number of eosinophils per HPF (*x*-axis) in each donor (dot). Linear regression lines are shown. FDRs were from two-tailed one-sample Student's *t* tests. **d** Distinctive cellular composition profiles for each condition. Each sample (left) and cell type (right) by the first two principal components (PCs, *x*- and *y*-axis) of the sample (biopsy)-cell type count profile matrix after centered log-ratio transformation. Source data are provided as a Source Data file.

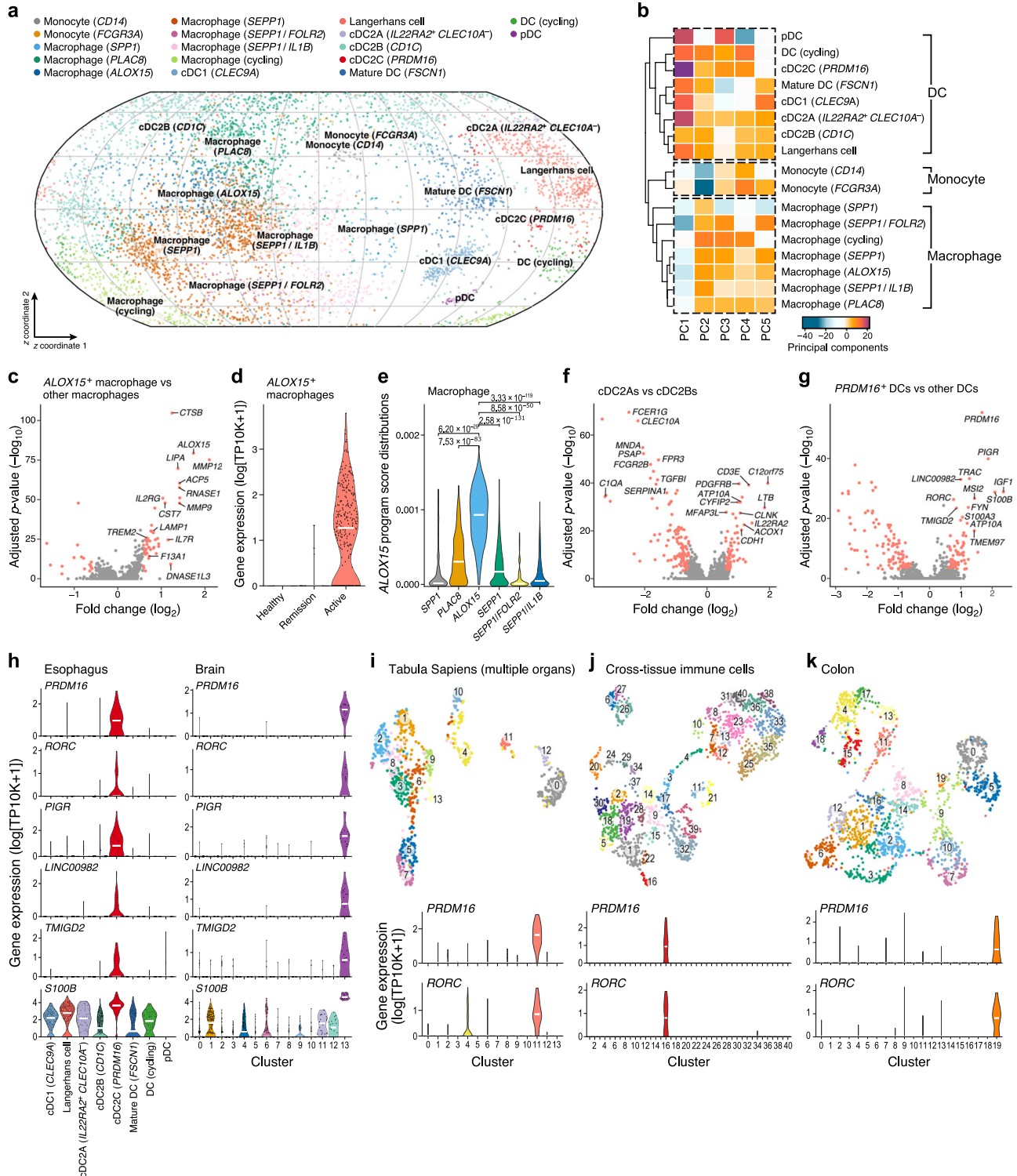

are a rare DC subset, increase in EoE, and express the EoE risk gene *ATP10A*.

### Rare resident ILC2s produce IL-13, IL-5, and prostaglandins

We identified rare ILC2s (Fig. 4a, Supplementary Fig. 2b, Supplementary Data 4) that were significantly increased in proportion in active EoE compared to health (Fig. 2a, Supplementary Fig. 3e). ILC2 are likely tissue-resident, based on expression of *CD69* and *ITGAE* but not the circulation marker *FAM65B*[45] (Fig. 4b). Consistent with tissue residency, esophageal ILC2s had higher expression of markers that are highly expressed in lung and colon ILC2s but low in blood and tonsil

ILC2s[46] (Fig. 4c). Both ILC2s and $T_H$2 cells expressed IL-13 and IL-5, but ILC2s also upregulated prostaglandin-related genes *PTGDR2, PTGS2*, and *HPGD* (Fig. 4d). Interestingly, ILC2s and $T_H$2 cells expressed phospholamban (*PLN*), which is induced by TGFβ1 in human esophageal smooth muscle cells and EoE myofibroblasts[47], but ILC2s in the esophagus expressed amphiregulin (*AREG*) at higher levels, suggesting that they may function in tissue repair during EoE[48].

### $T_H$17 cells decrease in active EoE and downregulate cytokines

While CD8[+] T cells and NK cells accounted for ~65% of the detected T and NK cells, they did not change in proportions in patients with active

**Fig. 3 | Disease-associated *ALOX15*⁺ macrophages and *PRDM16*⁺ DCs in the EoE myeloid compartment. a, b** Myeloid compartment of the esophageal cell atlas. **a** Two-dimensional spherical latent representation of 6515 monocytes, macrophages, and DC profiles (dots) from all donors ($n = 22$), colored by cell type, as learned by scPhere[82], taking 10x Chromium library version as the batch factor. **b** Scores (color) on each of the top 5 PCs (columns) for each myeloid cell subset (rows) hierarchically clustered based on Euclidean distance (dendrogram on left) with complete linkage. **c–e** *ALOX15A*⁺ macrophages in active EoE. **c** Significance (y-axis, -log₁₀(p-value), two-tailed likelihood-ratio tests of logistic regression coefficients, Bonferroni corrected) of differential expression (fold change, x-axis) for each gene (dot) between *ALOX15*⁺ macrophages and other macrophages. Red: >1.5-fold change, adjusted $p < 0.001$. **d** Distribution of expression (y-axis, log₁₀(TP10K + 1)) of *ALOX15* in *ALOX15*⁺ macrophages from each condition (x-axis). **e** Distribution of *ALOX15* NMF program scores (y-axis) in different macrophage subsets (x-axis). FDR (BH): two-tailed Mann–Whitney U test. **f** DC2s subsets. Significance (y-axis (−log₁₀(p-value), two-tailed likelihood-ratio tests of logistic

regression coefficients, Bonferroni corrected) of differential expression (fold change, x-axis) for each gene (dot) between cDC2A and cDC2B (classic *CD1C*⁺ cDC2). Red: >1.5-fold change, adjusted $p < 0.001$. **g–k** *PRDM16*⁺ cDC2Cs. **g** Significance (y-axis (−log₁₀(p-value), two-tailed likelihood-ratio tests of logistic regression coefficients, Bonferroni corrected) of differential expression (fold change, x-axis) for each gene (dot) between *PRDM16*⁺ cDC2Cs and other DCs. Red: >1.5-fold change, adjusted $p < 0.001$. **h** Distribution of expression (y-axis, log₁₀(TP10K + 1)) of cDC2C marker genes (x-axis) in cDC2Cs in our dataset (left) and from re-analyzing a publicly available brain dataset (right)[37]. **i–k** Top: UMAP Embedding of cell profiles from the multiple-organ Tabular Sapiens (**i**[42]), cross-tissue immune cells (**j**[43]), and colon (**k**[44]), colored by cluster. Bottom: Distribution of expression level (y-axis, log₁₀(TP10K + 1)) of *PRDM16*⁺ cDC2Cs marker genes in each cluster (x-axis). Violin plots: width based on Gaussian kernel density estimation of data with default parameters, scaled to a maximum of 1; White horizontal segment: median. Source data are provided as a Source Data file.

EoE, whereas CD4⁺ T_H cells showed changes, including in additionally characterized subsets (Fig. 2a, Supplementary Fig. 3e). Except for T_H2 cells, other CD4⁺ T_H cells have not been extensively characterized in EoE, and a recent scRNA-seq study of 1088 T cells in EoE reported only T_H2 cells and T_regs[13]. We identified these and five additional CD4⁺ T cell subsets with distinct markers, including T_CM cells expressing naive markers (*SELL* and *LEF1*) and *KLRB1*; T_RM cells expressing *CCL5, ITGA1*, and *TNF*; T_H1 cells expressing *IFNG, CXCR3*, and *CCL4*; T_H17 cells expressing *IL17A, IL22* and *CCR6*; and cycling T cells (Fig. 1d). Reanalysis of the 1088 profiles[13] with our signatures confirmed the presence of small numbers of T_CM and T_H17 cells in that dataset (Fig. 4e).

Compared to those in healthy participants, T_H17 cells were depleted in patients with active EoE in both the public data (FDR < 0.001, Fisher's exact test; Fig. 4e) and our data (Fig. 2a) and down-regulated *IL17A/F, IL22*, and *IL26* (FDR < 0.001, likelihood-ratio test, Fig. 4f). This is consistent with IL-17 depletion in peripheral blood mononuclear cells (PBMCs) from pediatric EoE patients[49]. Within T cells, T_H2 proportions were increased and T_H17 proportions were decreased in active EoE compared to healthy participants (FDR < 0.001, Wald test; Fig. 4g). Thus, EoE impacts the T_H cell compartment, while the composition of cytotoxic cell subsets is broadly unaffected.

### *IFNG*⁺ T cells and interferon response signatures correlate across cell types

Previous reports noted a dominant type 1 (IFN-γ) *vs.* type 2 (IL-4/5/13) cytokine production in esophageal T cells, and elevated interferon-α/γ response signatures in esophageal tissues[13,50]. Consistently, CD8⁺ T cells, CD4⁺ T_H1 cells, and *FCERG3*⁺ NK cells all expressed interferon-γ (*IFNG*), and most of the 60 cell subsets had elevated interferon-α and/ or interferon-γ response gene signatures (>0, Supplementary Fig. 4a, Methods). A few cell subsets, including epithelial cells, eosinophils, mast cells, and plasma B cells, did not show a high interferon-α/γ response signature.

Interestingly, for many subsets, the proportion of *IFNG*⁺ CD8⁺ T cells (out of all CD8⁺ T cells) was correlated across samples with the average interferon-γ response gene signature score of that subset (Supplementary Fig. 4b). This was especially notable for T_regs, CD4⁺ central memory T cells, CD8⁺ effector T cells, and MHC II CD8⁺ T cells (Supplementary Fig. 4b,c). Conversely, the expression of this signature in cycling pericytes was negatively correlated with *IFNG*⁺ CD8⁺ T cell proportions across samples. This analysis indicates that the presence of *IFNG*⁺ CD8⁺ T cells may dynamically alter gene expression across other T cell subsets.

### Diverse plasma B cells and IgE⁺ B cells increase in active EoE

An increase in plasma B cell proportions was a salient feature of active EoE (rising from a median of 0.021% of all cells in non-active EoE, including both healthy and remission, to 0.535% in active EoE),

especially for IgG⁺ (from a median of 0 to 0.044% across patients, FDR < 0.05, Wald test), IgM⁺ (from a median of 0 to 0.0019% across patients, FDR < 0.05), and cycling (from median of 0 to 0.024% across patients, FDR < 0.05) plasma cells, which represent only a handful (0–22) of cells detected in non-active EoE (Fig. 2a, b, d, Supplementary Fig. 3e, Supplementary Fig. 5a, b). Within the B cell compartment, cycling plasma B cell proportions increased from a median of 0.00% in non-active EoE to 3.60% in active EoE (FDR < 0.001, Wald test; Supplementary Fig. 5c). Notably, IgG⁺ plasma B cells expressed different subsets of IgG genes (Supplementary Fig. 5d, e), either solely *IGHG4*, co-expression of *IGHG2* and *IGHG1* (Spearman ρ = 0.53), or co-expression of *IGHG1* and *IGHG3* (minimum Spearman ρ = 0.70).

Both healthy and EoE samples contained a subset of memory B cells that expressed high levels of MHC-II genes and *CD40*[51] along with low levels of different immunoglobulin class genes, including *IGHE* (Supplementary Fig. 5a, f). IgE⁺ memory B cells are rare[52,53] and may not have been previously described in vivo in EoE. IgE-expressing memory and plasma B cells were absent in healthy participants (FDR < 0.001, Fisher's exact test; Supplementary Fig. 5g).

### Shared and cell type-specific gene program changes in active EoE

We next focused on cell-intrinsic changes in EoE by examining genes that were differentially expressed in cells of the same type between healthy, active EoE, and remission samples (Fig. 5a, Supplementary Fig. 6a–c). Across the 60 prevalent subsets, hundreds of genes in total were differentially expressed in specific cell types between active EoE and health (mostly from epithelial, stromal, or mast cells, with 133 differentially expressed genes detected from apical cells) or between active EoE and remission (Supplementary Fig. 6a, c). Fewer genes were differentially expressed in those cells between remission and health, indicating some renormalization of gene programs in remission.

Specifically, of the genes differentially expressed in cells between active EoE and healthy samples, 26.7% were also similarly differentially expressed (in the same cell type) between active and remission samples (Fig. 5a, top, Pearson $R = 0.69$, Supplementary Fig. 6b, c), consistent with normalization, and 22.3% were differentially expressed between remission and healthy samples, consistent with lingering differences (Fig. 5a, center, Pearson $R = 0.77$, Supplementary Fig. 6b, c). This suggests that while many changes normalize in remission, others do not, or do so only partly. Indeed, many of the genes differentially expressed in specific cell types between remission and health followed similar expression trends in active EoE (Fig. 5a, bottom, Pearson $R = 0.61$, Supplementary Fig. 6b, c). Genes with cell type-specific expression changes in active EoE that normalized in remission were enriched for immune functions and regulators of immune system processes, exocytosis, and peptidase activity, including multiple genes in fibroblasts, macrophages, and cDC2Bs (Fig. 5b,

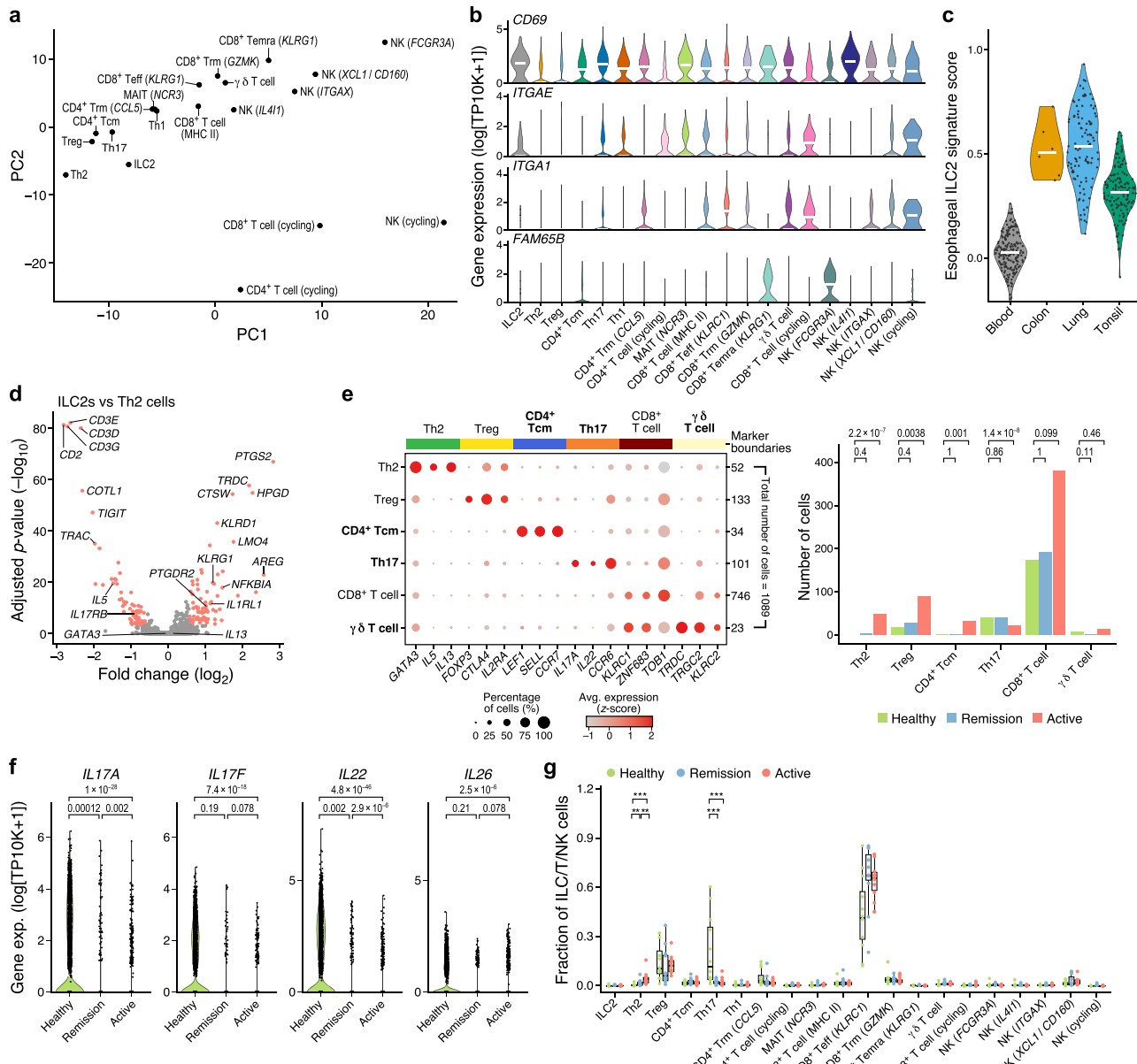

**Fig. 4 | Characterizations of ILC2s and T cells in EoE. a** T, NK and ILC cell compartment. Pseudobulk cell profiles of each cell subset (dots) in the space of the first two PCs of a PCA applied to the pseudobulk cell profiles. **b–d** Esophageal ILC2s express prostaglandin-related genes. **b** Distribution of expression (*y*-axis, $\log_{10}(\text{TP10K}+1)$) of tissue-resident (*CD69, ITGAE, ITGA1*) and circulating (*FAM65B*) marker gene across the subsets in the T/ILC/NK compartment (*x*-axis). **c** Distribution of scores (*y*-axis) of an esophageal ILC2 signature in ILC2s from different tissues (*x*-axis). **d** Significance (*y*-axis (−$\log_{10}$(*p*-value), two-tailed likelihood-ratio tests of logistic regression coefficients, Bonferroni corrected) of differential expression (fold change, *x*-axis) for each gene (dot) between ILC2s and T_H2 cells. Red: >1.5-fold change, adjusted *p* < 0.001. **e–g** Expansion of T_H2 cells, contraction of T_H17 cells, and reduction in T_H17 signature cytokines in EoE. **e** Left: Mean expression (dot color) and proportion of expressing cells (dot size) of marker genes (columns) of different T_H cell subsets (rows) in re-analysis of 1089 publicly available

T cell profiles from EoE patients[13]. Right: Proportion of cells (*y*-axis) of each T cell subset (*x*-axis) detected in each condition (bar color). ***FDR < 0.001, **FDR < 0.01, two-tailed Fisher's exact test. **f** Distribution of expression (*y*-axis, $\log_{10}(\text{TP10K}+1)$) of key cytokine genes in T_H17 cells from each condition (*x*-axis). FDRs of differential expression analysis (two-tailed likelihood-ratio tests of logistic regression coefficients) indicated on top. **g** Distribution of cell type proportions (*y*-axis) of each T/NK/ILC cell subset (*x*-axis) in each biopsy (dot) in each condition (color) (healthy: *n* = 12; remission: *n* = 11; active: *n* = 14). ***FDR < 0.001, **FDR < 0.01, *FDR < 0.05, two-tailed Wald test. Boxplots: medians and interquartile ranges (IQR). Whiskers: lowest datum within 1.5 IQR of the lower quartile and highest datum within 1.5 IQR of the upper quartile. Two-tailed Wald test FDRs are indicated on top. Violin plots: width based on Gaussian kernel density estimation of data with default parameters, scaled to a maximum of 1; White horizontal segment: median. Source data are provided as a Source Data file.

Supplementary Fig. 6a, Supplementary Data 5, Methods). Conversely, genes with expression changes that were sustained in remission were enriched for cornification, skin development, regulation of epithelial cell proliferation, response to lipid, and regulation of cell motility functions, and were mostly expressed in apical cells and pericytes (Supplementary Fig. 6d, Supplementary Data 5).

In particular, fibroblast genes dysregulated in active EoE and normalized fully or partly in remission were highly enriched for positive immune regulators (Fig. 5b), including *IL13RA2, IGF2, C3, CFD, IGFBP2, GAS6,* and *C1S* (Fig. 5c). *IL13RA2*, encoding a decoy receptor for IL-13 and a signature gene for inflammatory fibroblasts enriched in ulcerative colitis[54], was expressed in fibroblasts almost exclusively in

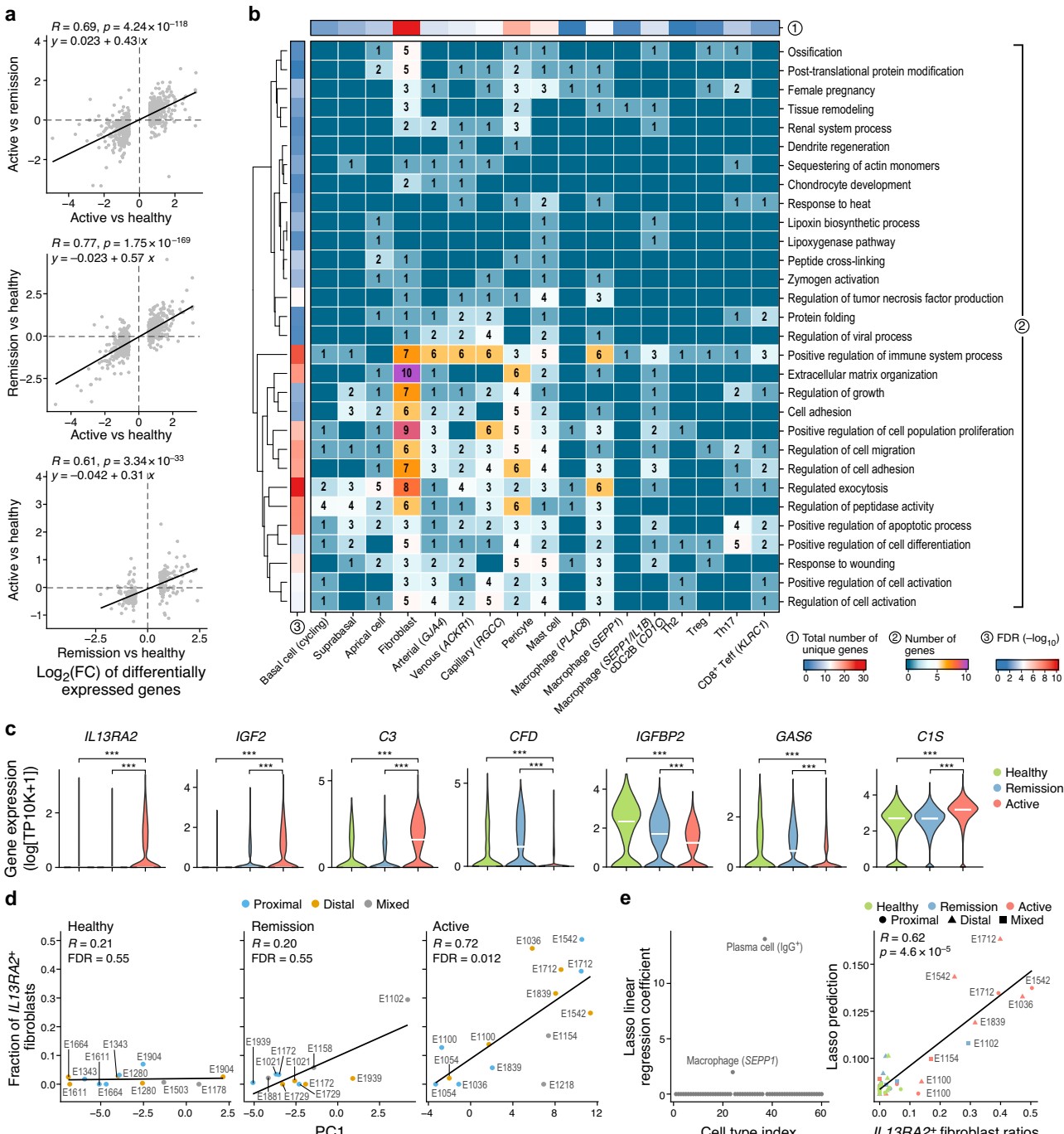

**Fig. 5 | Gene programs changes in active EoE and normalization in remission.**
**a**, **b** Gene changes in cell-intrinsic programs in EoE. **a** Genes (dots) with significant differential expression (log₂(fold changes), *x*- and *y*-axis) between different conditions (axis labels). Pearson correlation coefficients and linear regression lines shown in upper left corners. The *p*-values are based on two-tailed one-sample Student's *t* tests. **b** Enrichment (-log₁₀(FDR), one-tailed hypergeometric test) of Gene Ontology biological process terms (rows) in genes differentially expressed (numbers) between active EoE and both health and remission samples, in each cell type (columns). **c–e** Shift to *IL13RA2*⁺ inflammatory fibroblasts in active EoE.
**c** Distribution of expression (*y*-axis, log₁₀(TP10K + 1)) in fibroblasts from the three conditions (*x*-axis), of genes differentially expressed in fibroblasts in EoE *vs.* healthy and annotated as 'positive regulation of immune system process'. Width: Gaussian

kernel density estimation of data with default parameters, scaled to have a maximum of 1; White horizontal segment: median. ***FDR < 0.001, two-tailed likelihood-ratio tests of logistic regression coefficients. **d** The fraction of *IL13RA2*⁺ fibroblasts (*y*-axis) and PC1 score of the centered log-ratio transformed cellular composition profile (as in Fig. 2d) for each sample (dot) from healthy (left), remission (middle) or active EoE (right) patients. The FDRs are based on two-tailed one-sample Student's *t* test. **e** Left: Lasso linear regression coefficient (*y*-axis, left) of each cell type (dot) (ordered by cell type index (*x*-axis), as in Fig. 1d) as predictors of fraction of *IL13RA2*⁺ fibroblasts out of all fibroblasts. Right: Plots of lasso linear regression predictions (*y*-axis, based on IgG⁺ plasma cell ratios) and observed *IL13RA2*⁺ fibroblasts ratios. The black line is a fitted linear regression line. The *p*-values are based on two-tailed one-sample Student's *t* test. Source data are provided as a Source Data file.

active EoE (Fig. 5c). The proportion of *IL13RA2*⁺ fibroblasts out of all fibroblasts correlated with PC1 (Fig. 5d), which reflected disease state in a PCA of cell type proportions (Fig. 2d). This was driven by both low proportions of these cells in healthy participants and remission patients (Fig. 5d, left and middle, with one outlier) and by a strong positive correlation with PC1 in active EoE patients (Fig. 5d, right; Spearman $\rho = 0.72$). The proportion of *IL13RA2*⁺ fibroblasts was higher in distal biopsies and lower in matched proximal biopsies E1036 and E1839, suggesting a location-specific association (Fig. 5d). Lasso linear regression further associated the percentages of *IL13RA2*⁺ fibroblasts with IgG⁺ plasma cells and *SEPP1*⁺ macrophages (Fig. 5e, Methods). Broadly, these results demonstrate that EoE is also marked by cell type-specific intrinsic expression responses, including the expression of *IL13RA2* in fibroblasts as an important correlate of disease activity and changes in the esophageal mucosa.

## Multicellular interactions recruit immune cells in active EoE

We next focused on changes in inferred cell–cell interactions between conditions by connecting cells expressing ligands to those expressing cognate receptors in health, remission, and active EoE separately (Fig. 6a, Supplementary Fig. 7a, Methods). We also assessed if putative interacting cells co-varied in their cell proportions.

In active EoE, cells with the most interactions with eosinophils included fibroblasts, venous endothelial cells, T$_H$2 cells, suprabasal cells, and pericytes (Supplementary Fig. 7b); fibroblasts, venous endothelial cells, and pericytes also co-varied in their cell proportion (Fig. 6b, Methods). Previous studies showed that in patients with active EoE, fully mature eosinophils in the bone marrow are recruited to inflammatory sites through IL-5, IL-13, and eotaxins produced by immune, stromal, and epithelial cells[55–57]. In our atlas, activated BECs expressed adhesion molecules, including intercellular adhesion molecules (*ICAM1/2*), P-selectin (*SELP*), and *MADCAM1* (mucosal vascular addressing cell adhesion molecule 1) (Fig. 6c, Supplementary Fig. 7b), which bind to integrin α4β7 and L-selectin expressed on eosinophils upon stimulation by selective chemoattractants, such as eotaxins and *CCL5*[58]. Eosinophils expressed *CCR3*, the C-C chemokine receptor for eotaxin-1, and *CCL26* (eotaxin-3) (Fig. 6c, Supplementary Fig. 7b). *CCL26* was also expressed in pericytes and suprabasal cells, whereas *CCL11* (eotaxin-1) was mostly expressed in fibroblasts (Fig. 6c, Supplementary Fig. 7b). IL-5, the growth, activation, and survival factor for eosinophils, was expressed in ILC2s and T$_H$2 cells (Fig. 6c). These genes were highly expressed in active EoE compared to remission or health (Fig. 6d).

As expected, expanding mast cells in active EoE had increased interactions with several cell subsets, including mast cells themselves, endothelial cells, T$_H$2 cells, and fibroblasts (Supplementary Fig. 7a). Conversely, diminished T$_H$17 cells in active EoE decreased their interactions with epithelial cells (especially cycling basal cells, Supplementary Fig. 7a). Fibroblasts, which did not expand in active EoE compared to health or remission (Fig. 2a) had increased interactions with many cell states (Fig. 6a, Supplementary Fig. 7a), including fibroblasts themselves and venous endothelial cells (which did not expand in active EoE compared to health or remission, Fig. 2a). The top interactions between fibroblasts and other cell subsets were mediated by IL13–IL13RA2 (T$_H$2 cells), C3–C3AR1 (mast cells), CCL11–ACKR1 (venous endothelial cells), PDGFC–PDGFRA and FN1–ITGAV (fibroblasts), and VCAN–CD44 (cycling basal cells) (Supplementary Fig. 8a, b).

T$_H$2 cells expressing IL-5, IL-4, IL-13, and IL-9 had many putative interactions with mast cells expressing the cognate receptors (Fig. 6a, including the IL-9 receptor genes *IL9R* and *IL2RG*) and co-varied in cell proportions (Fig. 6b). *IL9* expression was only detected in active EoE (Supplementary Fig. 8c), and the IL-9–IL-9R ligand–receptor pair was among the top interactions between T$_H$2 cells and mast cells. IL-9 is a mast cell growth and survival factor, and eosinophils are a source of IL-

9 in the esophageal mucosa[59]. Mast cells also putatively interacted with BECs (Supplementary Fig. 7a) through IL-33–ST2, LIF–LIFR/LI6ST, and histamine receptor HRH1 (Fig. 6e). BECs are also involved in eosinophil recruitment (Fig. 6a, c, d, Supplementary Fig. 7b) and expressed the S1P receptor gene *S1PR1*. As BECs may interact with both mast cells and eosinophils, *S1PR1* may be a therapeutic target for EoE.

Apical cells both highly interacted and co-varied in proportions with quiescent basal cells (Fig. 6b, Supplementary Fig. 7a), and these interactions were higher in active EoE and remission compared to health. The top ligand–receptor pairs included IL1RN–IL1R2, EGFR–TGFA, and IL-36A–IL1RL2/IL1RAP (Supplementary Fig. 8d). Remarkably, apical cells expressed each of the 11 IL-1 family cytokine genes except for *IL1B* (expressed in myeloid cells), *IL33* (expressed in BECs and quiescent basal cells), and *IL37* (nearly undetectable) (Fig. 6f). Apical cells expressed the receptor antagonist IL-1RN that blocks IL-1 signaling (Fig. 6f, Supplementary Fig. 8d). Quiescent basal cells highly expressed *IL1R2* (Supplementary Fig. 8d), encoding a non-signaling receptor for IL-1A, IL-1B, and IL-1RN. Expression of the receptor antagonist and decoy receptor suggests that IL-1 signaling is tightly controlled in the esophageal epithelium. Taken together, our interaction analysis provides direct insights into potential cell–cell interaction networks underlying the compositional differences observed during disease and highlights potential therapeutic avenues for EoE, especially as many of these pathways have been targeted in other disease contexts.

## Using the esophageal cell atlas as a reference

Our esophageal cell atlas provides a detailed census to help interpret data from smaller, more focused studies. For example, we classified the 14,392 cells from pediatric EoE patients[18] based on our atlas' refined annotations (Supplementary Fig. 9a, b). Our atlas' larger size yielded richer annotations of rarer cell subsets within broader categories, and those in turn allowed us to identify cell profiles that were too scarce to be distinguished as a cluster and annotated as a subset in the original study. These included *ALOX15*⁺ macrophages, *PRDM16*⁺ DCs, ILC2s, *FCN1*⁺ monocytes (originally assigned to a granulocyte subset), and *MS4A1*⁺ memory B cells (originally included in a myeloid subset) (Supplementary Fig. 9a, b).

We also used the atlas as a reference to refine the source cells of key genes. The pediatric EoE study highlighted the expression of the GPR15 ligand gene *C10ORF99* in epithelial cells[18], and our finer mapping identified suprabasal cells as the major source cell in both datasets (Supplementary Fig. 9b, c). This demonstrates the utility of a large-scale atlas in providing cellular context and annotations to anchor existing or future studies of the esophageal mucosa.

## EoE risk gene expression highlights an apical cell-specific module

Finally, we analyzed the cell type-specific expression of 52 genes associated with EoE risk by genome-wide association studies (GWAS)[60,61], candidate gene association studies[62], expression quantitative trait loci (eQTL) analysis[63], or exon sequencing[64] and in Mendelian diseases associated with EoE (hyper-IgE syndrome, Ehlers-Danlos syndrome, PTEN hamartoma tumor syndrome, ERBIN deficiency, Loeys-Dietz syndrome, SAM syndrome, and Netherton's syndrome)[65] (Methods).

Of the nine causal genes of Mendelian diseases associated with EoE, four (*STAT3*, *PTEN*, *ERBB2IP*, and *TGFBR2*) were expressed in stromal cells and myeloid cells, two (*DOCK8* and *TGFBR1*) were also frequently expressed in myeloid cells, two (*DSP* and *SPINK5*) were mostly expressed in epithelial cells, and one (*DSG1*) was weakly expressed (Supplementary Fig. 10a, Methods).

Of the other putative EoE risk genes, 56.8% were expressed in one or more cell subsets. Epithelial cells, especially apical cells, expressed *CAPN5/14*, *FLG*, and *SHROOM3*. *CAPN5/14* and *SHROOM3* expression

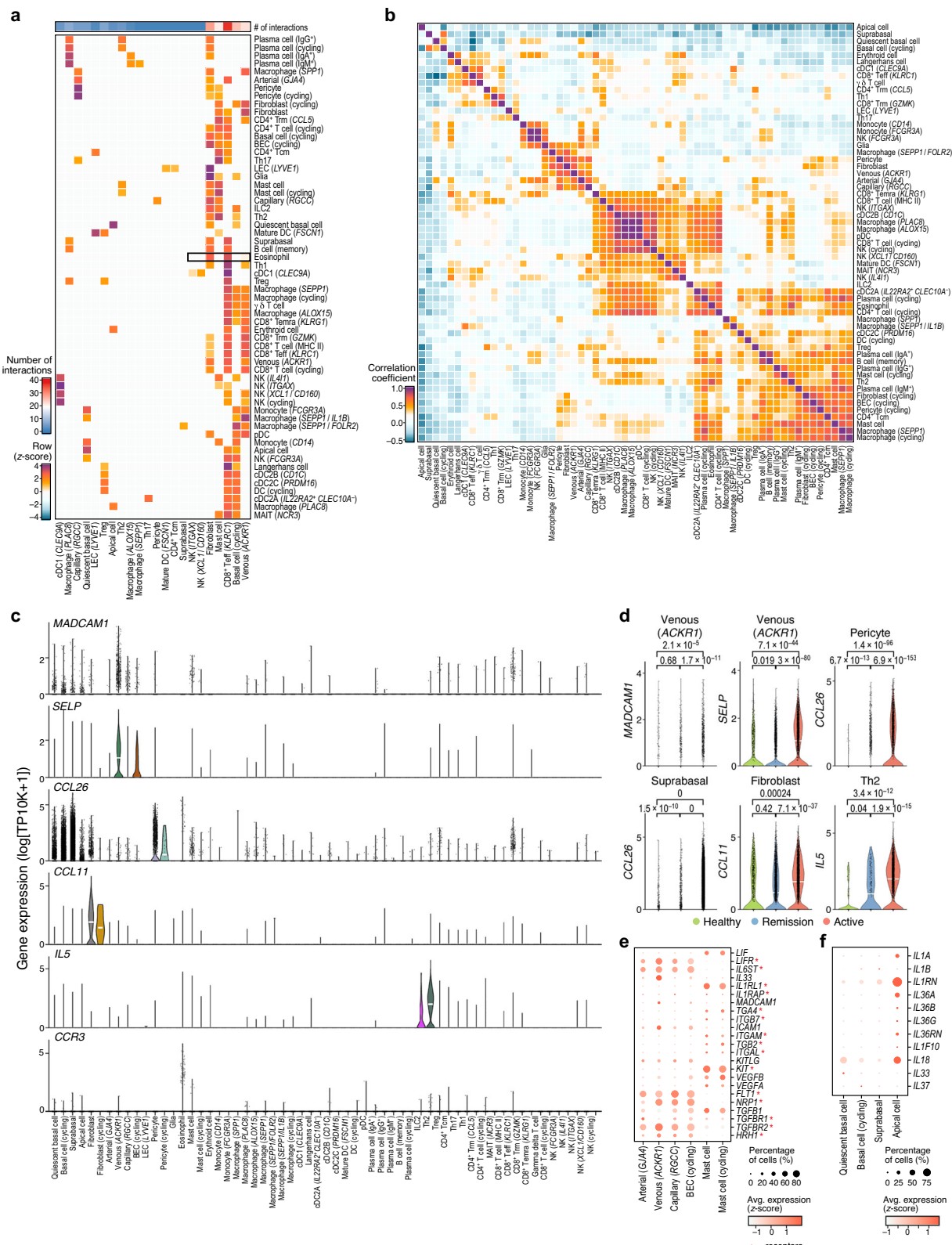

Nature Communications | (2024)15:3344 11

also correlated across apical cells (minimum Spearman ρ = 0.52, Supplementary Fig. 10b), indicating a potential apical cell-specific contribution to EoE pathophysiology. Other putative EoE risk genes were specifically expressed in quiescent basal cells (*TSLP*), fibroblasts (*FBN1* and *NOVA1*), BECs (*SEMA6A* and *LRRC32*), pericytes (*CCL26*), mast cells (*IL1RL1*), ILC2s and T_H2 cells (*GATA3* and *KIF3A*), and cDC2As and cDC2Cs (*ATP10A*). While certain cycling cell subsets (e.g., fibroblasts,

BECs, plasma cells, and NK cells) expressed more risk genes than the corresponding non-cycling subsets, this may be due to higher RNA content and scRNA-seq complexity (Supplementary Fig. 10a).

The Mendelian and putative EoE risk genes formed modules based on their cell type-specific associations (Supplementary Fig. 10c) and co-expression (Supplementary Fig. 10d), some capturing both Mendelian and common risk (GWAS) genes in the same module. We

**Fig. 6 | Cells communicate to maintain tissue homeostasis and coordinate to recruit immune cells in EoE. a** Putative ligand–receptor interactions. Ligand–receptor interaction weights (color, row Z-scores) for the top 3 interaction partners (columns) for each cell type (rows). Top row: number of interactions across all types. **b** Correlated cell types. Pearson's correlation coefficient (color bar) between the proportion of each pair of cell types (rows, columns) across samples ($n = 37$) clustered by hierarchical clustering with the Mcquitty linkage method. **c** Eosinophil recruitment genes are expressed by activated venous endothelial cells, pericytes/suprabasal cells/fibroblasts, ILC2s and $T_H2$ cells and induced in EoE. Distribution of expression (y-axis, $\log_{10}(TP10K + 1)$) in each cell type (**c**, x-axis) or in specific cell types across conditions (**d**), of genes encoding key eosinophil recruitment factors. Width: Gaussian kernel density estimation of data with default parameters, scaled to have a maximum of 1; White horizontal segment: median. FDR of differential expression (two-tailed likelihood-ratio test) indicated on top. **e** Mast and blood endothelial cell interactions. Mean expression (dot color) and proportion of expressing cells (dot size) in mast and endothelial cell subsets (columns) of genes encoding receptor (asterisk) and ligand pairs expressed by these cells. **f** Apical cells from active EoE express most of the IL-1 family cytokine genes and receptor antagonists. Mean expression (dot color) and proportion of expressing cells (dot size) in epithelial cell subsets (columns) of genes encoding IL1 family cytokines and receptor antagonists (rows). Source data are provided as a Source Data file.

identified modules of apical cell-specific genes (GWAS genes *SHROOM3*, *CAPN5*, *CAPN14*, and *FLG* and Mendelian genes *DSP* and *SPINK5*), of $T_H2$- and ILC2-specific genes (GWAS genes *GATA3* and *KIF3A*), of cycling BEC-specific genes (GWAS genes *LRRC32* and *WDR36*), and a module associated with multiple immune cells (Mendelian genes *TGFBR1* and *PTEN* and GWAS genes *STAT6* and *SLC25A24*) (Supplementary Fig. 10c, d).

## Discussion

Limited knowledge about the human esophagus has hindered efforts to understand the underlying nature of highly tissue- and organ-specific diseases such as EoE. We applied single-cell profiling of human esophageal biopsies to comprehensively describe the cellular landscape of the esophagus in health and inflammation. Partitioning single-cell profiles into 60 prevalent subsets with distinct marker genes highlighted the cellular diversity and immunologic complexity of the esophagus, including 11 cycling cell subsets, generally expanded in active EoE, and several unconventional MHC-II-expressing cell types.

Our data suggest an important role for mast cells in EoE pathogenesis. In addition to the expected increase in eosinophils, which were almost exclusively recovered from biopsies collected from patients with active EoE, cycling mast cells were also specific to active EoE in our data. Our cell–cell interaction analysis suggests that mast cells may act through direct interactions with BECs, which is consistent with observations of decreased mucosal impedance associated with active EoE[66]. Notably, mast cells highly expressed the EoE risk gene *IL1RL1*, also expressed in the rare ILC2s (Supplementary Fig. 10a). Moreover, we identified $T_H2$ cells as an additional source of the mast cell growth factor IL-9 in active EoE. Sustained mast cell-mediated inflammation may explain why earlier trials of IL-5 blockers mepolizumab and reslizumab reduced eosinophil counts in the esophagus, but did not induce histologic remission or significant improvement in symptoms, and would also be consistent with reports of EoE-like disease without tissue eosinophilia, but with mast cell infiltration[67], and with the description of a subset of EoE patients who have epithelium-restricted mast cell infiltration, despite resolution of epithelial eosinophilia[68]. In contrast to IL-5 blockade, dupilumab, which targets the IL-4 receptor alpha subunit required for both IL-4 and IL-13 signaling, is a promising therapeutic approach for a subset of individuals with EoE[69], potentially due to its broader effects on non-eosinophil cells involved in type 2 inflammatory diseases, including suppressive effects on both $T_H2$ cell differentiation as well as inhibition of inflammation mediated by $T_H2$- and ILC2-derived IL-13.

Our atlas allowed us to study macrophages and DCs, which are relatively rare (1.63% of all detected cells) and have not been widely studied in the human esophagus but express multiple EoE risk genes. *ALOX15*⁺ macrophages and *PRDM16*⁺ cDC2Cs increased in frequency in active EoE. *ALOX15* encodes a 15-lipoxygenase whose metabolites promote polarization of macrophages to an M2 phenotype[34]; higher esophageal expression of *ALOX15* has been described in refractory or relapsing patients[70], whereas downregulation of *ALOX15* expression characterizes mild EoE with a normal-appearing esophagus[71]. Because

*ALOX15* expression depends on IL-4 and IL-13, it has been suggested that dupilumab may counter type 2 inflammation in chronic rhinosinusitis in part by suppressing an IL-4/IL-13/ALOX15 M2 macrophage signaling axis[72,73]. Our data thus suggest an underappreciated role for esophageal macrophages in EoE pathogenesis. To our knowledge, *PRDM16*⁺ cDC2Cs have not been characterized before. These cells had expression profiles similar to *TBX21*⁺*RORC* cDC2A cells recently identified in the spleen[40] and may be cDC2 progenitor cells[74], but further work is required to map their fate commitments. Both cDC2As and cDC2Cs specifically expressed *ATP10A*, suggesting a potential cell-type specific role for this EoE risk gene in these DC subsets. Further studies delineating the function of the additionally annotated cDC2Cs and the role of *ATP10A* in this subset will be of high interest.

High-resolution scRNA-seq also elucidated a relationship between IgG4-expressing plasma cells and a relatively recently described subset of inflammatory fibroblasts[54]. The role of IgG4 in EoE is incompletely understood, but previous studies demonstrated that tissue IgG4 levels correlate closely with esophageal eosinophil counts, histologic severity, and levels of esophageal IL-4 and IL-13[17]. Additional studies have described abundant IgG4-containing plasma cells[16] and elevated food-specific IgG4 levels in the blood. *IL13RA2*⁺ inflammatory fibroblasts, which also increase in inflammatory bowel disease[54], were almost undetectable in healthy participants but expanded in some active EoE patients. In active EoE, the percentage of *IL13RA2*⁺ inflammatory fibroblasts relative to all fibroblasts correlated with the proportions of IgG⁺ plasma B cells and *SEPP1*⁺ macrophages. This association between IgG4⁺ plasma cells and *IL13RA2*⁺ inflammatory fibroblasts in EoE further supports IgG4 as a potential mediator of disease, and suggests a further role for plasma cells, in concert with inflammatory fibroblasts, in promoting inflammation and histologic injury in EoE.

ILC2s, which are very rare in the human esophagus[10], highly expressed genes related to the production of prostaglandins, which are lipid mediators that participate in diverse homeostatic processes in many tissue types. Prostaglandin D2 (PGD2) and E2 (PGE2) play an important role in the function of multiple cell types involved in allergic immune responses, including epithelial barrier cells, DCs, macrophages, mast cells, and eosinophils, where PGD2 generally enhances eosinopoiesis, chemotaxis, integrin expression, degranulation, and survival[75]. Little is known about the role in EoE of prostaglandin-endoperoxide synthase 2 (*PTGS2*), which is upstream of PGD2 synthesis. One study examining only epithelial tissue showed reduced levels of *PTGS2* in EoE patients compared to healthy participants or patients with reflux esophagitis[76]. We detected ILCs in the lamina propria[54,77] and described the specific upregulation in EoE of prostaglandin-related gene expression in this rare immune cell population.

$T_H17$ cells and apical epithelial cells were reduced in proportions in active EoE, with $T_H17$ cells also expressing lower levels of *IL17A/F* and *IL22* transcripts, potentially due to suppression of $T_H17$ cell differentiation and cytokine production by eosinophils[78]. In our cell–cell interaction analysis, $T_H17$ cells putatively interacted with epithelial cells in healthy participants, which may promote barrier integrity

under homeostatic conditions[79]. Interestingly, the loss of $T_H17$ cell and marker cytokine expression was sustained in patients in remission, suggesting that these cells may not be recovered as type 2 inflammation subsides. This is reminiscent of the long-term loss of a subset of intra-epithelial γδ T cells in celiac disease[80], but needs to be substantiated in the case of EoE through extended follow-up.

Utilizing EoE as a robust model of type 2 cytokine-driven mucosal inflammation, our study provides a framework for applying scRNA-seq to understand how type 2 immune responses mobilize multiple cell types and signaling pathways. Moreover, our extensive and deeply annotated cell atlas of the human esophagus in health and inflammation provides a valuable resource for rapidly identifying sequenced cells from the esophageal mucosa for broader investigations of specific cell types and their interactions in this previously understudied segment of the gastrointestinal tract, as well as for comparative analyses with different tissues and tissue diseases.

## Methods

### Patients and tissue samples

The study protocol complied with all relevant ethics regulations and was approved by the Mass General Brigham Institutional Review Board (study ID: 2015P000816). Human esophageal endoscopic biopsies were obtained from 15 patients with eosinophilic esophagitis (EoE) and 7 healthy participants after informed written consent. Clinical data and demographics (including patient sex, age range at sample collection, disease status, number of eosinophils per high power field, and diet and treatment information) are provided in Supplementary Data 1. We recruited the same number of male ($n = 11$) and female ($n = 11$) participants. Given the limited number of participants within each phenotype and that sex is convoluted with the 10x library preparation method (see below), no sex-specific analysis results were reported in this study.

### Sample collection and single-cell dissociation

Biopsies were collected from 22 donors: 7 healthy participants, 8 patients with active disease, and 7 patients in remission. For 15 donors, biopsies were obtained from both the distal and the proximal regions separately.

Biopsies were transported to the Broad Institute in Advanced DMEM/F-12 (Gibco, cat. no. 12634028). To maintain cell quality, tissue dissociation was initiated within 3 h of biopsy collection. Prior to sample arrival, enzymatic digestion media was freshly prepared by combining 50 ml of cell isolation media ($CO_2$ independent media (Invitrogen, cat. no. 18045088) supplemented with 2% FBS, 1% non-essential amino acids and 1% GlutaMax) with 100 µl Collagenase (100 mg/ml, Roche, cat. no. 11249002001, reconstituted in sterile HBSS), 100 µl Dispase (100 mg/ml, Gibco, cat. no. 17105041, reconstituted in sterile HBSS) and 100 µl DNAse I (50 mg/ml, Roche, cat. no. 10104159001, reconstituted in 20 mM Tris HCl, 1 mM MgCl2, 50% glycerol in HBSS).

Enzymatic digestion media was kept at room temperature throughout the experiment. Biopsies were washed using cold PBS and transferred to a sterile 50 ml conical tubes containing 5 ml of enzymatic digestion media. Each tube was vortexed for 30 s and placed in a water bath at 37 °C for 5 min. After 5 min, tubes were removed from the water bath and vortexed again for 30 s each. This was repeated two additional times for a total incubation time of 15 min. Tissue was allowed to settle to the bottom of the conical tube, supernatant was carefully collected using a P1000 and moved into a sterile 50 ml conical tube on ice containing 15 ml cold isolation media supplemented with 5 mM EDTA.

5 ml enzymatic digestion media was added to each of the remaining biopsies, and samples were vortexed and incubated for an additional 10 min, with vortexing at 5 min. The supernatant was collected from the digest and added to the previously collected supernatant. 2 ml enzymatic digestion media was added to each of the remaining biopsies, mixed using a P1000. Samples were incubated in the water bath for 3 min, mixed forcefully, and returned to the water bath for 2 additional minutes. Supernatant was collected and added to previously collected supernatants. If visible tissue remained, an additional 10–15 min of digestion was performed in the water bath, vortexing for 30 s after each 5-minute period, and supernatant was collected as described.

The supernatant and media mixture was filtered through a 100 µm cell strainer (Falcon, cat. no. 352360) into a 50 ml conical tube and topped up to 50 ml with cell isolation media. Cells were centrifuged at 300 g for 10 min at 4 °C. Supernatant was removed and discarded, leaving around 0.5-1 ml of supernatant. Cells were resuspended and transferred to a 1.7 ml Eppendorf tube and pelleted using the short spin centrifuge setting with centrifugal force ramping up to 11,000 g. Supernatant was removed and cells were resuspended in 50–300 µl of PBS/0.01% BSA depending on pellet size. The supernatant was discarded and cells were resuspended in 50–300 µl of PBS/0.01% BSA, depending on pellet size.

To count cells and assess viability, 5 µl of cell suspension was mixed with 5 µl Trypan Blue (Thermo Fisher Scientific, cat. no. T10282) and loaded onto an INCYTO C-Chip Disposable Hemocytometer, Neubauer Improved (VWR, cat. no. 82030-468). Cells were counted manually and, if necessary, diluted to a concentration of 200–2000 cells/µl.

### Single-cell RNA-seq

For each sample, 4 channels (Chromium v3.1 dual-index chemistry data) or 8 channels of 8000 cells each were loaded on a 10x Genomics Single-Cell Chromium Controller.

For seven of the patients, 10x Chromium v2 chemistry was used for library preparation and libraries were sequenced on an Illumina HiSeq with the following read configuration: R1 (cell barcode and UMI): 26 bp, i7 index: 8 bp, R2 (insert): 98 bp.

For 9 patients, NextGEM v3.1 single-index 3' Chemistry was used and libraries were sequenced on an Illumina HiSeq with the following read configuration: R1 (cell barcode and UMI): 28 bp, i7 index: 8 bp, R2 (insert): 96 bp.

For the remaining 6 patients, NextGEM v3.1 dual-index 3' Chemistry was used and libraries were sequenced on an Illumina HiSeq with the following read configuration: R1 (cell barcode and UMI): 28 bp, i7 index: 10 bp, i5 index: 10 bp, R2 (insert): 90 bp.

### Single-cell RNA-seq data pre-processing

CellRanger-2.1.1 (for 10x Chromium v2 chemistry data), CellRanger-3.1.0 (for Chromium v3.1 single-index chemistry data), and CellRanger-5.0.0 (for Chromium v3.1 dual-index chemistry data) were used for read demultiplexing, alignment to the human GRCh38 genome (from CellRanger refdata v1.2.0), and unique molecular identifier (UMI) counting and collapsing. Cell profiles with more than 500 UMIs and less than 25% UMIs from mitochondrial transcripts were retained for downstream analysis for v2 chemistry data. For v3 chemistry data, a higher threshold of 40% mitochondrial transcripts was used, as these data are known to typically have higher proportions of mitochondrial transcripts[81].

### Single-cell RNA-seq data integration, clustering, and annotation

scPhere v0.1.0[82], a deep learning-based method, was used to integrate all cell profiles by taking patient origin of cells, disease status, and spatial locations of biopsies as batch vectors (Supplementary Data 1). Although the version of the 10x Genomics chemistry is a potential confounding factor correlated with gene expression, this information was not taken as a batch vector because (1) the version was convolved in our study with biopsy spatial information, and (2) the more granular patient information was already used as a batch vector

(Supplementary Data 1). A latent space of 10 dimensions was used, such that each cell was mapped to a 10-sphere and the latent representation of cells was then clustered using the Louvain community detection algorithm[83,84] to produce 30 clusters. These clusters were merged (assigned) to 23 putative cell types/groups using an automatic cell type assignment approach, as previously described[81], followed by manual inspection.

To identify cell subtypes (e.g., different macrophage/DC subtypes), we re-clustered the cells of interest, (e.g., macrophages/DCs) using either scPhere embeddings or the R package Seurat v4.1.1. (R version v4.2.0, all statistical tests were based on this version of R if not specified). Cell clusters that expressed marker genes of two cell types were labeled as 'contaminants' (e.g., pericyte clusters that also expressed epithelial cell marker genes) and clusters that had very small numbers of detected genes per cell and did not express cell subtype-specific marker genes were labeled as 'low UMI'. The contaminants/doublets included a cluster of BEC/pericyte doublets (with 3865 mean detected genes vs. 2056 and 1235 for BECs and pericytes, respectively), which are likely biological doublets of physically interacting pericytes wrapped tightly around blood capillary endothelial[85]. Notably, the ratio between BEC/pericyte doublet numbers and pericyte numbers was 77.60% (3737/4817), much higher than the expected doublet rate (-5% for 10x Chromium data). These contaminants/doublets and low UMI cells were removed from further analysis. DensityCut v0.01 was used to identify very rare cell states, combined with visualization[86]. Overall, 20T/natural killer/lymphoid cell subsets, 17 monocyte/macrophage/dendritic cell subsets, 10 stromal (endothelial, pericyte, fibroblast, and glial) cell subsets, 5 B cell subsets, 4 epithelial cell subsets, 3 granulocyte (mast and eosinophil) cell subsets, and 1 red blood (erythroid) cell subset were retained for further analysis.

## Visualizing the structure in high-dimensional scRNA-seq data
To visualize the high-dimensional scRNA-seq data, cells were embedded on a sphere using scPhere v0.1.0, or in a 2-dimensional UMAP[26]. For efficient exploratory data analysis, a pseudobulk profile was generated for cell profiles in each subset (see Pseudo-bulk analysis below), followed by principal component analysis (PCA).

## Permutational multivariate analysis of variance (PERMANOVA)
PERMANOVA was performed to show the effects of disease, diet elimination, steroid, use of proton pump inhibitors (PPIs), and age on cellular composition using the vegan package[87]. The PERMANOVA procedure calculates an F-statistic from the original data and F-statistics from multiple permuted data to calculate p-values. The number of permutations was set to 1 million. To perform PERMANOVA, a distance matrix between samples (biopsies) was generated, where each sample was represented by its cellular composition, quantile normalized across samples, and the Euclidean distance was used as the distance between samples. We did an extra analysis by also adding patient sex, and the results were almost unchanged. Of note, 6/7 donor samples used for 10x Chromium v2 chemistry library preparation were male, and 7/9 donor samples used for 10x v3 single-index library preparation were female. As 10x library and sex were convoluted, we only added 10x library version as a covariate for analysis and did not include sex for differential expression analysis and cell compositional shift analysis.

## Statistical significance of shifts in cell composition
To assess the statistical significance of changes in cell composition, a negative binomial regression model was used as previously described[77,88]. First, disease (active EoE, remission EoE, or healthy), version (10x version 2, version 3, or version 3 dual index), treatment (steroid), and region (distal, proximal, or mixed) were used as covariates, and the total number of analyzed cells from each biopsy was used

as an offset variable. The significance of disease (active EoE) on a cell type was assessed using the Wald test on the regression coefficients, followed by False Discovery Rate (FDR) estimate using the Benjamini-Hochberg (BH) method. Because an increase in the proportion of one cell type results in decreases in all other cell types in a sample, compositional analysis was performed to further explore cellular compositional variations in different disease conditions by transforming the sample-cell type count matrix based on the centered log-ratio transformation after imputing zero counts[89]. The transformed data were used for PCA to visualize and help interpret cellular compositional shifts in disease conditions (Fig. 2d).

In addition to modeling counts using negative binomial regression, cell type proportions were also calculated, which are bounded between 0 and 1. Cell type proportion distributions across samples (biopsies) can be different for different cell types (e.g., with higher variance for rare cell types). Assuming that most cell types do not change cellular composition in EoE, the cell type proportions were quantile normalized before quantifying cellular compositional changes. After quantile normalization, for each cell state, linear regression was used to model the ranks of cellular compositions, by taking disease, version, steroid, and region as predictors. Significant tests of individual regression coefficients were obtained by one sample t-tests.

## Differential expression analysis
Logistic regression was used for differential expression analysis, taking both the $\log_2$-transformed total number of detected genes in each cell, treatment, region, and the 10x Chromium library version as covariates, using the Seurat v4.1.1 implementation for differential expression analysis[90].

To filter out differentially expressed genes that were likely due to ambient RNA, each gene $g$ of a cell type $c$ was tested for upregulation in each of the $M$ samples (e.g., $M = 14$ active EoE biopsies when comparing active EoE to healthy samples). If $g$ was downregulated in most of the $M$ samples, it was removed from further analysis. The STRINGdb[91] Bioconductor package (using STRING v11) was used for Gene Ontology enrichment analysis with the 'get_enrichment' function for enrichment analysis, and all the genes in CellRanger refdata v1.2.0, after removing mitochondrial genes and ribosome protein coding genes (i.e., RPS* and RPL* genes), used as the background gene list.

## Lasso linear regression to predict IL13RA2$^+$ fibroblast percentages
To select cell composition features that predict the percentages of IL13RA2$^+$ fibroblasts (relative to all fibroblasts in a biopsy), $L1$-regularized linear regression (Lasso) was used with features of ratio of a cell type in a biopsy (60 features, 37 biopsies). Leave-one-out cross-validation was performed to select the optimal penalty hyper-parameter for Lasso regression, and the selected hyper-parameter was used for feature selection and training a linear regression model.

## Pseudo-bulk analysis
Pseudo-bulk data analysis was performed as previously described[81]. A pseudo-bulk expression profile for a set of cells was computed by taking the sum of the cells' expression vectors (each a raw UMI count vector) where the vectors were normalized by dividing the total number of UMIs from all cells, multiplying by $10^4$, and finally taking the natural log (adding one before the log transformation to make all the elements of the bulk vector positive).

## Gene signature analysis
ILC2 gene signatures were obtained from an scRNA-seq study of ILC2s[46] and gene signature scores were calculated as previously described[92], as implemented in the Seurat v4.1.1 package[90]. The interferon-α and interferon-γ response gene signatures were

downloaded from the Molecular Signature Database (MSigDB)[93]. Non-negative matrix factorization was applied to all macrophage profiles to identify gene programs[94] and their scores in each cell.

## Cell–cell interaction analysis

A set of 585 manually curated ligand–receptor pairs was used to identify putative cell-cell interactions, as previously described[95]. Ligand–receptor pairs included multi-subunit complexes such as *ITGA4*/*ITGB7*. Current methods for inferring cell–cell interactions from scRNA-seq face several challenges. First, if a ligand or receptor is ubiquitously highly expressed, cell types with the cognate receptor or ligand can be designated as hubs. Second, some methods ignore multi-subunit complexes. Third, noise in scRNA-seq can lead to spurious putative interactions, and those can lead to erroneous overall signal given the large number of ligand–receptor pairs used in ranking the interaction strength between two cell types.

To address some of the challenges, a $k$-nearest neighbor approach was used to estimate the interaction strength between cell types. Specifically, the score of a gene $g$ in a cell subset $s_g$ is defined as the total number of detected UMIs of gene $g$ in that cell subset divided by the total number of UMIs of gene $g$ across all cell subsets. If $g$ is a member of a multi-subunit complex with $m$ subunits, the weight for that complex is defined as the geometric mean of scores of all members. Then the final interaction score of a ligand–receptor pair between two cell subsets is the product of the weights of the two interaction complexes in the two cell subsets. For any two cell types $i$ and $j$, the interaction score is calculated for each ligand–receptor pair. Finally, the interaction strength between two cell types is the sum of the top $k$ interaction scores. These interaction strength scores are used to rank the interaction partners of a cell type. Here, $k = 10$ was used to decrease the influence of the background interaction noise. When using $k = 5$ or 15, the top interactions partners for each cell type were largely unchanged, indicating the robustness of the results to this parameter choice[96]. We analyzed the biopsies from healthy, remission EoE, and active EoE participants separately.

## Annotation of public scRNA-seq data with the esophageal cell atlas

To map the 14,392 cells from the esophageal mucosa of pediatric EoE patients[18] with the esophageal cell atlas in our study, despite batch effects between and within studies, neighborhood component analysis[97] was used to map the reference to a 50-dimensional latent space, and then the learned neighborhood component analysis mapping function was used to project the public data to the same 50-dimensional latent space. Next a $k$-nearest neighbor classifier ($k = 11$) trained on our data was used to predict cell identities of the test data (all in the 50-dimensional latent space).

## EoE risk gene analysis

A set of 52 EoE risk genes was compiled from recent reviews[60,98], including those from candidate gene or GWAS[23,62,99], eQTL analysis[63], and whole-exome sequencing[64], as well as genes for Mendelian diseases with EoE-like phenotypes[65].

Genes were considered for further analysis if they were expressed in at least 25% of cells of a subset (Supplementary Fig. 10a,c). Pseudo-bulk analysis was used for detecting modules from EoE risk genes. In contrast to other pseudo-bulk procedures, we used mean instead of sum in aggregating the counts of a gene across cells of a state. We further did hierarchical clustering with the 'Ward.D2' lineage function on the square root of the pseudo-bulk to detect modules.

## Reporting summary

Further information on research design is available in the Nature Portfolio Reporting Summary linked to this article.

## Data availability

The raw FASTQ files of the scRNA-seq data generated in this study are available in the database for Genotypes and Phenotypes (dbGAP) under accession code phs003574.v1.p1 (http://www.ncbi.nlm.nih.gov/projects/gap/cgi-bin/study.cgi?study_id=phs003574.v1.p1). These data are available under restricted access to prevent misuse of patient genetic information. Access can be obtained by submitting an online Data Access Request (DAR) through the dbGAP Authorized Access page (https://dbgap.ncbi.nlm.nih.gov/aa/wga.cgi?page=login). The processed gene count matrices and embeddings of the scRNA-seq data generated in this study are available at https://singlecell.broadinstitute.org/single_cell/study/SCP1242/eoe-eosinophilic-esophagitis. The 1088 T cell dataset used in this study is available in GEO under accession GSE126250. The pediatric EoE dataset used in this study is available in GEO under accession GSE175930. The glioblastoma dataset used in this study is available from the Brain Immune Atlas (https://www.brainimmuneatlas.org/download.php). The colon dataset used in this study is available at https://www.gutcellatlas.org. The cross-tissue immune cell dataset used in this study is available at https://www.tissueimmunecellatlas.org. The Tabula Sapiens dataset used in this study is available at https://tabula-sapiens-portal.ds.czbiohub.org. The GRCh38 human reference 1.2.0 used in this study was downloaded from 10x Genomics (https://www.10xgenomics.com/support/software/cell-ranger/downloads) and can be regenerated according to the instructions at https://www.10xgenomics.com/support/software/cell-ranger/downloads/cr-ref-build-steps. All other data are available in the article and its Supplementary files or from the corresponding author upon request. Source data are provided with this paper.

## Code availability

The code for this project is available in GitHub at https://github.com/Ding-Group/eoe.

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

## Acknowledgements

The authors are grateful to the patients who participated in this study. We thank Theresa Reimels for manuscript edits and comments; Leslie Gaffney for help with figure preparation; Phylicia Dorceus, Corey Nolet, and Dan Dubinsky for scRNA-seq experiments; and Yinian Niu for help analyzing public scRNA-seq data. This project was supported by grants from the Food Allergy Science Initiative (to R.J.X. and A.R.), Klarman Cell Observatory at Broad Institute, National Institutes of Health (P30 DK043351 and RC2 DK114784 to R.J.X.), and a Discovery grant from the Natural Sciences and Engineering Research Council (NSERC) of Canada (to J.Ding). J.Ding is a Canada Research Chair.

## Author contributions

J.J.G., R.J.X., A.R., and D.B.G. conceived and designed the study. A.L., G.T.C., T.M.D., O.R.-R., and J.Deguine performed scRNA-seq experiments. A.U., J.J.G., P.V., L.C., M.D., K.S., and J.Y. performed clinical work. J.Ding, A.R., D.B.G., and R.J.X. designed and performed computational analyses and data interpretation. O.A. helped with data analysis. J.Ding, J.J.G., J.Deguine, A.R., and R.J.X. wrote the manuscript with input from all authors.

## Competing interests

R.J.X. is a co-founder of Jnana Therapeutics and Celsius Therapeutics, a scientific advisory board member at Nestlé, and board director at MoonLake Immunotherapeutics; these organizations had no roles in this study. A.R. is a founder and equity holder of Celsius Therapeutics, is an equity holder in Immunitas Therapeutics, and, until 31 August 2020, was a scientific advisory board member of Syros Pharmaceuticals, Neogene Therapeutics, Asimov, and ThermoFisher Scientific. Since 1 August 2020, A.R. has been an employee of Genentech, a member of the Roche Group, with equity in Roche. K.S. is a consultant to Sanofi. All other authors declare no competing interests.
