## [Peer Review File · Nature Communications]

An esophagus cell atlas reveals dynamic rewiring during active eosinophilic esophagitis and remissionEditorial Note: This manuscript has been previously reviewed at another journal that is not operating a transparent peer review scheme. This document only contains reviewer comments and rebuttal letters for versions considered at *Nature Communications*.

REVIEWERS' COMMENTS

Reviewer #2 (Remarks to the Author):

The authors have addressed most of my previous concerns. I remain underwhelmed by the biological insights derived from this work but, as I mentioned in my original comments, I think that the data set is in itself an important contribution to the field.

The only thing that I still find a little unclear/misleading from the paper is the description of cell type-specific gene program alterations in active EoE. The authors now show in a supplementary figure the number of genes classified as differentially expressed (DE) in each cell type. That figure shows very small numbers of DE genes per cell type but in the text the authors say "Across the 60 prevalent subsets, hundreds of genes were differentially expressed". Although that statement is true it gives the wrong impression that active EoE is associated with a very large remodeling of the transcriptome, which is not an exact description of the data.

Furthermore, there is a discrepancy I can't reconcile: the minimal DE genes shown in Supplementary Figure 6A and the incredibly low p-values (below 10^{-300}) presented in Supplementary Figure 6B. How can these two observations be consistent?

Reviewer #3 (Remarks to the Author):

The manuscript by Ding and colleagues describes large-scale transcriptomics profiling of 421,312 individual cells from the esophageal mucosa of healthy (n = 7) and Eosinophilic Esophagitis EoE (n = 15) subjects. Prior concerns related to clinical significance due to deficiencies in clinical descriptors (symptoms and therapy) of the patient cohorts as well as relationship between gene expression and cell types / states with endoscopic and histologic characteristics and disease endotypes.

In general, the authors have done a commendable job at addressing all these concerns and the manuscript while descriptive provides biological insight into EoE-associated cellular features and will provide the EoE research community with an invaluable resource for new discoveries.

Reviewer #4 (Remarks to the Author):

The authors have adequately addressed the comments raised by Reviewer #1. Of note, additional metadata for all patients was provided, along with several additional analyses on specific cellular populations.

Reviewers' Comments

Reviewer #2 (Remarks to the Author):

The authors have addressed most of my previous concerns. I remain underwhelmed by the biological insights derived from this work but, as I mentioned in my original comments, I think that the data set is in itself an important contribution to the field.

The only thing that I still find a little unclear/misleading from the paper is the description of cell type-specific gene program alterations in active EoE. The authors now show in a supplementary figure the number of genes classified as differentially expressed (DE) in each cell type. That figure shows very small numbers of DE genes per cell type but in the text the authors say "Across the 60 prevalent subsets, hundreds of genes were differentially expressed". Although that statement is true it gives the wrong impression that active EoE is associated with a very large remodeling of the transcriptome, which is not an exact description of the data.

Furthermore, there is a discrepancy I can't reconcile: the minimal DE genes shown in Supplementary Figure 6A and the incredibly low p -values (below 10^{-300}) presented in Supplementary Figure 6B. How can these two observations be consistent?

We thank the reviewer for the comments that helped us improve the clarity and precision of our descriptions. We revised the sentence as follows:

“Across the 60 prevalent subsets, hundreds of genes in total were differentially expressed in specific cell types between active EoE and health (mostly from epithelial, stromal, or mast cells, with 133 differentially expressed genes detected from apical cells) or between active EoE and remission (Supplementary Fig. 6a,c).”

For our differentially expressed (DE) gene analysis, we used logistic regression implemented in the Seurat package. Although these tools are the community standard for differential expression testing, they consider each cell as an independent observation, but the cells are from a few patients. Thus, typically they produce small p -values. As for the small number of DE genes, one of the reasons is because we filtered DE genes that were likely to be contaminated by ambient RNAs. Please refer to our methods section about differential expression analysis for detail, also reproduced below for your convenience:

“To filter out differentially expressed genes that were likely due to ambient RNA, each gene g of a cell type c was tested for upregulation in each of the M samples (*e.g.*, $M=14$ active EoE biopsies when comparing active EoE to healthy samples). If g was downregulated in most of the M samples, it was removed from further analysis.”

Reviewer #3 (Remarks to the Author):

The manuscript by Ding and colleagues describes large-scale transcriptomics profiling of 421,312 individual cells from the esophageal mucosa of healthy (n = 7) and Eosinophilic Esophagitis EoE (n = 15) subjects. Prior concerns related to clinical significance due to deficiencies in clinical descriptors (symptoms and therapy) of the patient cohorts as well as relationship between gene expression and cell types / states with endoscopic and histologic characteristics and disease endotypes.

In general, the authors have done a commendable job at addressing all these concerns and the manuscript while descriptive provides biological insight into EoE-associated cellular features and will provide the EoE research community with an invaluable resource for new discoveries.

We appreciate the reviewer for all the comments that helped us improve our manuscript. We are very glad to hear that the reviewer believes our manuscript and data provide biological insight into EoE and an invaluable resource for the community.

Reviewer #4 (Remarks to the Author):

The authors have adequately addressed the comments raised by Reviewer #1. Of note, additional metadata for all patients was provided, along with several additional analyses on specific cellular populations.

We appreciate the reviewer for considering that our revision has adequately addressed the comments raised by Reviewer #1.